# The Developmental Origins of Health and Disease: Adolescence as a Critical Lifecourse Period to Break the Transgenerational Cycle of NCDs—A Narrative Review

**DOI:** 10.3390/ijerph19106024

**Published:** 2022-05-16

**Authors:** Melenaite Tohi, Jacquie Lindsay Bay, Siobhan Tu’akoi, Mark Hedley Vickers

**Affiliations:** 1Liggins Institute, The University of Auckland, Auckland 1023, New Zealand; melenaite.tohi@auckland.ac.nz (M.T.); j.bay@auckland.ac.nz (J.L.B.); 2School of Population Health, The University of Auckland, Auckland 1023, New Zealand; s.tuakoi@auckland.ac.nz

**Keywords:** developmental origins of health and disease, DOHaD, noncommunicable diseases, developmental programming, adolescents, adolescence, lifecourse

## Abstract

Noncommunicable diseases (NCDs), including type 2 diabetes and cardiovascular disease, represent a significant and growing global health burden. To date, a primary focus has been on treatment approaches to NCDs once manifested rather than strategies aimed at prevention. In this context, there is clear evidence that a range of adverse early life exposures can predispose individuals towards a greater risk of developing NCDs across the lifecourse. These risk factors can be passed to future generations, thus perpetuating a cycle of disease. This concept, preferentially termed “developmental programming”, forms the basis of the Developmental Origins of Health and Disease (DOHaD) framework. To date, DOHaD has focused on preconception, pregnancy, lactation and, more recently, paternal health at the time of conception. However, it is becoming increasingly clear that investment in the window of adolescence is perhaps the most critical developmental window. Adolescence is a period where lifestyle behaviours become entrained. Therefore, a focus on adolescent behaviours, health literacy and emotional development may afford the best opportunity to break the cycle of NCDs. As the next generation of parents, adolescents should therefore be considered a priority group in advancing appropriate and informed actions aimed at reducing NCD risk factors across the lifecourse. This advancement requires a more comprehensive community understanding and uptake of DOHaD knowledge and concepts. NCD prevention strategies have typically entailed siloed (and often disease-specific) approaches with limited efficacy in curbing NCD prevalence and breaking the transgenerational transmission of disease traits. Recent findings across various disciplines have highlighted that a lifecourse systems approach is required to establish a comprehensive and sustainable framework for NCD intervention. A whole community approach with a particular focus on adolescents as potential agents of change is necessary to break the disease cycle.

## 1. Introduction

It is well established that noncommunicable diseases (NCDs) are the leading cause of mortality worldwide [1]. According to the World Health Organization (WHO), NCDs accounted for 73.6% of all deaths in 2019 [2]. More than 15 million people between the ages of 30 and 69 die prematurely from NCDs annually [2], 85% of which occur in low- and middle-income countries. Premature deaths occurring during adulthood are often the result of health-related behaviours initiated in childhood and adolescence [3]. NCDs include type 2 diabetes (T2DM), cardiovascular diseases, some cancers, and respiratory diseases. These disease clusters are responsible for more than three-quarters of deaths from NCDs [4]. The COVID-19 pandemic has highlighted the impact of NCDs. Disease severity, hospitalisations, and mortality rates significantly increase in those with NCDs or related predisposing factors [5]. This evidence further highlights the need to prioritise primary prevention strategies [6] as current approaches for treatment of disease once already manifest have limited efficacy and do not serve to address the potential transgenerational impacts underlying such diseases [7]. Since the Ottawa Charter in 1986, a number of initiatives have been implemented that have worked towards advancing NCD management and control. The Workplace Health Promotion (WHP) programmes targeting physical inactivity and unhealthy dietary habits [8] were one such major initiative. Other movements included community-based NCD prevention projects to build the capacity of policy-makers and promote healthy lifestyles throughout the lifespan [9] and the Countrywide Integrated Non-Communicable Diseases Intervention programme implemented in many European countries [10]. A partnership between WHO and the United Nations Educational, Scientific and Cultural Organization launched the initiative “Making Every School a Health Promoting School” to promote positive development and healthy behaviours such as physical activity, fitness, recreation and play, balanced nutrition, and prevention of tobacco use [11]. However, although there has been some progress over the years, this research area remains significantly limited [12].

This review focusses on cardiometabolic diseases and diabetes as these represent some of the most prevalent NCDs globally [13]. We highlight the role of adolescence as a life stage in which health promotion through knowledge translation has the potential to facilitate improved long-term and health outcomes. The novelty of this review rests on the idea that as studies continue to indicate the importance of health optimization during preconception, pregnancy, and in the early years of life [14], primary prevention of NCDs need to be focused more during childhood and adolescence (greater emphasis on latter) where the likelihood of reduced disease risk is greater.

## 2. Core Concept of the Developmental Origins of Health and Disease (DOHaD) Hypothesis: Focus on the Health of Mother and Offspring

There is no doubt that the early life environment can increase the risk of developing a range of NCDs in later life, effects that can be passed to future generations (see Figure 1). This concept is preferentially termed developmental programming and forms the basis of the developmental origins of health and disease (DOHaD) framework. The fetal and early neonatal periods of growth and development are characterised by periods of developmental plasticity and, as such, are sensitive to environmental cues [15,16]. During this period of plasticity, the fetus can make adaptive changes (predictive adaptive responses or PARS) based on maternal cues in anticipation of its expected postnatal environment [17]. An example is maternal malnutrition. If maternal nutritional intake is poor during preconception and pregnancy, the fetus will receive signals that the environment that it is about to enter is likely to be unfavourable. In response to these maternal cues, the fetus adapts by altering its metabolism and making tissue-specific adaptations to help prepare for survival in a suboptimal nutritional environment following birth [18], thus developing a “thrifty phenotype” [19]. However, although advantageous for short term survival, these adaptations may predispose to later disease risk when there is a “mismatch” between the anticipated and actual postnatal environments, e.g., a postnatal obesogenic environment [20].

Growing epidemiological evidence from the DOHaD field has increased focus on reducing the burden of a poor nutritional environment in both the mother and offspring [21,22]. This focus has included attempts at mainstreaming strategies for better interventions to address the lasting impacts of undernutrition in both mothers and offspring [23,24]. Given the rapidly rising prevalence of NCDs globally, there has been considerable attention given to the DOHaD framework as a lifecourse approach that can contribute to breaking the disease cycle [25,26] (see Figure 1).

Early DOHaD evidence demonstrated the link between birth weight and increased risk of developing NCDs in later life and acknowledged the socio-economic environment as a significant risk factor [16]. Although initial research focussed on maternal undernutrition and low birth weight, there has been growing interest in the effects of maternal overweight/obesity. There is clear evidence linking a maternal obesogenic environment with later life overweight/obesity and metabolic disorders in offspring. Of note, these effects have been shown experimentally to be independent of the postnatal dietary environment, thus representing a direct programming effect [27,28]. As an example, the Raine Study in Western Australia found that obesity and early mid-gestational weight gain in mothers were associated with nonalcoholic fatty liver disease in offspring [29]. Body mass index and weight gain of mothers before and in early pregnancy were associated with adiposity and cardiometabolic characteristics in their adolescent children [30]. In terms of the perinatal environment, stress induced by events during pregnancy such as divorce, pregnancy difficulties, financial problems, and the loss of a relative increased the risk for atopic disorders, asthma, and eczema in offspring [31].

Although poor maternal nutrition and stress is now well recognised as increasing the risk for a range of NCDs in offspring in later life, strategies to improve outcomes have met with limited success. Intervention studies attempted in the early phase of DOHaD research yielded little grounds for optimism. Growth enhancement in offspring when reproductive women were provided with optimal nutrition [32,33] showed no significant difference as collated by meta-analysis data from numerous randomised controlled trials (RCTs). More than a decade ago, nutrition supplementation interventions during pregnancy resulted in increased offspring birthweight through to no effect or even increased prevalence of low birth weigh [34,35,36,37]. These results suggest that focusing on pregnancy alone is difficult. The complex requirements around the maintenance of the energy/protein balance across pregnancy and dietary advice appears to unlikely confer significant benefits on maternal or infant health and, in some cases, may even be harmful [34].

Other behavioural factors such as maternal tobacco use, alcohol consumption, and stress [38] also contribute to increased risks of developing NCDs and high mortality rates [21,39,40,41,42]. Studies have shown that maternal smoking during pregnancy increased the risk of overweight or obesity for offspring in childhood [43]. Alarmingly, children who were exposed to maternal smoking had a higher risk of later being overweight [44] with higher systolic and diastolic blood pressure found in children whose mothers smoked during pregnancy [45]. This is of concern as overweight children typically track into overweight/obesity in adulthood [46]. Smoking is the most policy-responsive behavioural risk factor in public health. This is reflected in the tobacco control success in a number of high- and middle-income countries [40]. On the other hand, curbing of harmful levels of alcohol consumption in some Western countries remains a significant public health burden and appears to be worsening in Eastern Europe [47].

## 3. Influences of Paternal Health Behaviours on Long-Term Health and NCDs

While an extensive body of DOHaD research focuses on the health and well-being of the mother and offspring and the importance of early-life nutrition, the influence of paternal health on the growth and development of offspring has been overlooked [48]. Evidence around paternal factors and NCD risk factors in progeny has primarily been derived from experimental models [49,50], whereas epidemiological evidence around the transmission of disease traits from father to offspring is emerging [51,52,53]. This awareness of paternal effects supports recognizing the need to adopt a DOHaD lifecourse approach to address the growing NCD burden [54]. Further, the marked shift in NCD prevalence across different demographics further highlights the need for such a systems approach. Diseases such as T2DM are appearing in increasing numbers in children and adolescents, with the acknowledgement that NCDs no longer discriminate amongst age or gender [1,55,56]. It has often been assumed that adulthood is when the crucial risk factors affecting vulnerability operate and therefore has been a primary focus of interventions. However, this largely ignores the extensive evidence that suggests the early life developmental exposures can modify the responsiveness to later life exposures and lifestyle factors such as an obesogenic dietary environment and physical inactivity (see Figure 1). Further, given the increasing knowledge around both maternal and paternal factors on offspring outcomes and potential transgenerational impacts, there is an increasing need to translate DOHaD knowledge across a wider lifecourse perspective, including the period of adolescence, which influences the health of as future parents (Figure 1).

## 4. Knowledge Translation Approaches

Despite the traction gained by DOHaD-related research since the International DOHaD Society was formally established in 2003, the wider impact of DOHaD has been relatively slight [57]. DOHaD concepts are recognised by several Non-governmental Organisations (NGOs) but the relevance and contribution to these concepts to later disease is overlooked. This highlights the importance of knowledge being translated into practice to encourage evidence-based action. Knowledge translation (KT) involves the synthesis, dissemination and application of knowledge from research, practice, community experience, and culture to collectively inform actions [58]. Several WHO initiatives [59,60,61] have focused on the optimisation of pregnancy outcomes through addressing DOHaD concepts, including recommendations from groups such as the International Federation of Obstetricians and Gynecologists (FIGO) Working Group on Adolescent, Preconception and Maternal Nutrition. Other initiatives have prioritised the lifecourse approach to healthy aging [62,63,64] emphasising that addressing childhood obesity needs a lifecourse approach, focussing on conditions in early life that can lead to aberrant developmental programming and amplification of disease risk across the lifecourse due to an obesogenic environment [65].

One example of a current DOHaD KT approach is the dissemination of a nutrition resource aimed at pregnant women that shares evidence-based guidelines on nutrition throughout preconception, pregnancy, and toddlerhood [66]. This resource, titled the ‘First 1000 Days: Nutrition matters for lifelong health’, is distributed to new and expecting mothers in Australia and New Zealand. It is intended to promote awareness of the importance of early-life nutrition and encouraging the optimisation of their child’s lifelong health, thus reducing the risk factors associated with later-life diseases. Exploration of first time New Zealand mothers’ perceptions of the booklet identified the overwhelming pressure to comply with the list of recommendations, which resulted in feelings of shame, guilt, and a desire for more support from healthcare providers and society [67]. This evidence highlights the need for DOHaD knowledge to be made available in resources that can be used by health professionals working alongside women and their partners to support healthy child development [68].

Health promotion is pivotal for DOHaD KT approaches. Health promotion, as defined by Ottawa Charter in 1986, is the process of enabling people to increase control over and to improve their health [6]. Health promotion has proven to be effective in the sense that it has become a complementary framework to the traditional focus on health protection and disease prevention [13]. Health promotion through school-based diet and physical activity intervention programs continue to emerge as an important strategy for obesity prevention during the adolescence period in a number of countries [69,70,71,72,73,74,75]. Studies found that physical activity interventions were successful at increasing physical activity and lowering BMI in adolescents. Moreover, combined interventions of diet, physical activity, BMI, and prevalence of overweight or obesity showed significant outcomes [76]. These interventions need to be implemented in large groups of children and sustained over longer periods to explore the effects of these interventions on BMI [77]. However, rapid change in BMI over periods of school-based interventions is considered inappropriate [78]. The success of health promoting interventions has indicated that support from community, society, physical environment, and infrastructures have potential to influence the health of the adolescents in these settings [79] even though motivating and sustaining parental and family involvement remains a challenge in some countries [80,81]. Guided by empowerment, community participation, partnerships with multi-sectorial forums, and supportive environments [82], going forward we can expect that maximum benefit in diseases prevention will be achieved. Thus, a change in behaviour patterns regarding NCD prevention will filtrate from an individual to their community and society. Ultimately, we would expect the effects of successful health promotion to percolate from this generation to the next [13].

## 5. Adolescence: A Window of Opportunity for Better Health and Reduced NCD Risk

Cognitive, psychosocial, and lifestyle behaviours during adolescence develop and persist into adulthood [83]. As the next generation of parents, empowering adolescents affords the opportunity to contribute towards breaking the NCD risk and disease cycle. Trajectories of obesity are initiated at younger ages with birth cohort data, revealing that recent birth cohorts are becoming obese in greater proportions for a given age; therefore, individuals are experiencing a greater duration of obesity over their lifetime [84,85]. These rising obesity rates in childhood and adolescence increase the risk of later life NCD carried by individuals as they move into adulthood [86]. Adolescence represents a period where the individual gains increased freedom and agency around dietary and lifestyle choices and includes a period of experimentation often associated with risk behaviours such as smoking and excessive alcohol consumption. These behaviours tend to track into adulthood [87]. Thus, adolescence is a sensitive window of opportunity where optimisation of health and well-being can potentially act as a “circuit breaker” for transmission of NCD risk [88].

As NCDs develop slowly over time, NCD risk development also builds throughout the lifecourse. Given that the age at which NCDs are emerging is reducing, the importance of early life risk mitigation cannot be underestimated. Risk factors such as smoking, excessive alcohol consumption, lack of physical activity and unhealthy diets are becoming harder to manage in modern environments [89]. Thus, the impact of environmental exposures in all stages of life will contribute to the onset and long-term severity of NCDs. High-risk populations are often affected by high rates of unplanned pregnancies, many of which occur in younger mothers [90]. The magnitude of teenage pregnancy in developing nations is high [91], with most adolescent pregnancies likely to occur in those least developed countries [92].

Encouraging healthful behaviours, such as physical activity and healthy nutrition during childhood and adolescence, can improve later pregnancy outcomes, particularly in younger mothers [93]. Interventions that focus on the early postnatal period, i.e., the First 1000 Days, have met with limited success in the setting of unplanned pregnancies. Strategies need to be in place before these life stages to avoid reaching the point whereby the ‘horse has already bolted’ to change for a better future.

Although schools hold much potential for influencing healthy lifestyle behaviours in primary school age groups (5–11 years) [94], strategies to impact health behaviours at this age may also be constrained due to parental influences. Thus, a system-based approach that builds from pregnancy to adolescence represents a more integrated perspective to NCD prevention interventions.

Adolescents know their importance in yielding great dividends for future generations [95]. Adolescent DOHaD KT and health promotion work has been carried out in countries such as New Zealand [96], Japan [97], Tonga [98], Uganda [99], United Kingdom [100] and The Cook Islands [101]. Some of the earliest efforts to facilitate DOHaD KT via schools were implemented through LENScience, an ongoing programme that was launched in New Zealand in 2006 [102,103]. LENScience programmes utilise co-design to promote educational and science/health goals, ensuring valid and ethical use of curriculum-based learning time [104]. DOHaD evidence is reimaged and integrated into age-appropriate learning resources that link to core goals in the national curricula of the country of implementation [102]. These resources are used in learning programmes that engage narrative-based pedagogies to support the development of science and health literacy and promote learners to identify and experiment with evidence-based actions [78,102,105] (see Figure 2).

A number of adolescent-focussed DOHaD KT programmes have adapted and built on this narrative-based pedagogy to support DOHaD KT via curriculum-linked opportunities in areas such as science and health. Opportunities for school-based health promotion are also commonly explored via the internationally recognised Health Promoting Schools (HPS) movement [106]. In Uganda, DOHaD KT has been included in programs related to Health Promoting Schools (HPS). This was done through the introduction of the DOHaD concepts to 151 pupils aged 12–15 years in three HPS programs in rural Uganda. Students identified factors that they believed would make DOHaD-related health promotion resonate with them, and discussed how learning about DOHaD could be made acceptable to young people aged 12–15 [107,108]. The tertiary sector is another place where DOHaD KT opportunities have been identified. In Japan and New Zealand, exploration of DOHaD evidence has been intentionally placed within undergraduate programmes to promote increased awareness. However, evaluation of these efforts indicates that time dedicated to these concepts within crowded curricula should increase for meaningful KT to occur with undergraduate students [97]. This evaluation was carried out through the insertion of this model in the university curriculum for the relevant health professionals.

Evidence emerging from these existing DOHaD KT programmes indicates that adolescents are aware of how their health can contribute to ensuring a better health outcome. One critical step that appears to be under-developed involves looking at multiple ways to engage the wider community in understanding the importance of health during the adolescence for the current generation and their future offspring. Achieving this should involve multiple pathways and will need to be framed within the context of complex systems thinking. One area of interest is associated with the idea that the promotion of adolescent health and well-being can be advanced when presented by individuals or groups from within the demographic in question. By involving adolescents in the decision making, they are encouraged to contribute to breaking the cycle of NCDs [109]. The wider community can support adolescents by enabling them to voice their opinions on what aspects of their lives should be promoted.

The P45 Youth CAN (Change.Activity.Nutrition) program led by research students from Colorado State University is one example of an intervention that empowered adolescents to become agents of change for health promotion in their community [109]. This program highlighted the potential of youth-driven health initiatives for community changes that promote healthy eating and active living in Globeville-Elyria-Swansea (GES). The adolescents identified facilitators and barriers to healthy eating and physical activity through multiple mediums of Photovoice (photography), spoken word (poetry), and street art (graffiti-style artwork). Adolescents presented their findings to peers and community agencies through discussion using the World Café methodology [110], which generated numerous project ideas. Although there are still some steps to be taken before they reach the point where action plans are designed and implemented to promote health in GES, the adolescents are taking a stand and making a change. This change may seem small but has the potential to yield better health outcomes for the community in GES. Recognising that a positive social network for adolescents is a key factor in enhancing the health of the community contributes hugely to the change [111].

Use of the World Café method of deliberation has been shown to be effective in a range of contexts with adolescents and young adults, including those related to reproductive health [112]. The World Café model of deliberation promotes exploration of multiple perspectives. This could be an effective method to support adolescent-voices to co-lead the design and implementation of actions to promote understanding of the value of adolescent health for current and future generations.

## 6. Systems Approaches

An integrated approach that recognises that human health is a complex system is required to address the growing burden of NCDs. A systems approach looks holistically at the DOHaD lifecourse. As such, there is an increasing need to focus on the full potential of the DOHaD paradigm giving primary consideration to the pre- and periconceptional period (health behaviours, personal, societal, and environmental factors influencing behaviours before pregnancy and parenthood) [83]. An integrated approach recognises multiple intersecting pathways. Achieving this requires increasing knowledge and uptake of DOHaD concepts, from health and educational professionals right through to community leaders, youths, and adolescents as the next generation of parents [113]. Adolescents aged 10 to 24 years [114] occupy a significant stage/phase of the lifecourse, thus proving their greater relevance for human development.

The idea that “nutrition knowledge does not necessarily lead to behavioural change” has evolved considerably over the past two decades [115,116,117]. Research shows that the processes used to facilitate delivery of knowledge and the potential for people to develop understanding of the knowledge is crucial for successful interventions that focus on behavioural changes [118]. KT offers opportunities to improve health and nutritional literacy that can contribute to reducing later NCD risks and improve health outcomes for the next generation [119]. Success in health promotion interventions is associated with health literacy. Health literacy describes a range of outcomes of health education and communication activities [120]. It is more profound than just providing access to information or one’s ability to read and understand pamphlets or prescriptions (see Figure 3).

The promotion of capabilities that support health literacy holds a crucial role in addressing social, economic, and environmental determinants of health. Literature highlights that health literacy and scientific literacy work hand in hand to produce successful outcomes in health education [102]. However, a disconnect between health and scientific literacy currently exists [102]. The struggle to control the rise of obesity and related chronic diseases that often leads to the development of NCDs requires health literacy at all stages of life, from preconception and prenatal care right through to older people [121]. Moreover, the need for health literacy to be extended beyond the health sector through multi-sectorial work has been identified as a fundamental goal that could help reduce premature deaths by 30% [55]. Several studies have provided evidence of functional health literacy impacting the odds of adolescents engaging in risky health behaviours [122,123,124]. Interventions to promote health literacy in adults living with NCDs [125] and adolescents [95] have been progressing well in many high income countries, but little is known about the progress and impact of health literacy interventions in low-to-middle income countries. Follow-up studies need to be carried out in these settings to assess the level of knowledge and effectiveness of these interventions over time. The goal for health literacy is to support the development of capabilities that enable individuals and communities to access, interpret, and decide how to use evidence and knowledge to support informed decision-making about health. Outcomes resulting from skills developed through gained knowledge on an individual level includes improved knowledge of risk and health services and the capacity to act independently on that knowledge. This often increases the participation in population health programs at a community level, acting on social and economic determinants of health that eventually feed into evidence of progress in health literacy in the society, as in Figure 3. The translation of evidence into practice is vital as it can shift societal thinking through public communication. To date, DOHaD knowledge has been translated into action through the use of public communication strategies, such as the distribution of the early life nutrition resource detailed above, recently reimaged and contextualised for use in the Cook Islands [126] where NCD-related risk factors affect >80% of the adult population. It is considered progress within the NCD prevention movement when adolescents are given the opportunity to learn and explore what their overall health profile data means. In addition, it is often discouraging for a patient to be told they have high cholesterol without explanation, but it can be very empowering when health professionals and/or health promoters show evidence and explain what the numbers mean. This level of education and knowledge translation is what can encourage young people to make healthy changes and be the catalyst for change.

Although the translation of evidence is vital at every life stage, it is particularly crucial during adolescence. At this transitional phase between childhood and adulthood, young people begin to make their own decisions related to health and lifestyle behaviours [55]. We have reached a point where it makes so much sense to ask, “what about adolescents?” Why wait for the First 1000 Days when we can take steps to improve outcomes before then?

## 7. Conclusions

This narrative review highlights the need to focus on prevention rather than treatment to break the cycle of NCDs. Although the DOHaD paradigm initially focused on the early life origins of NCDs related to mothers, offspring, and the First 1000 days, recent research has extended it to include the impacts of paternal health. There is an increasing awareness that NCDs such as T2DM are no longer diseases of adulthood and are tracking from increasingly younger age groups. This has necessitated the drive to improve health literacy in DOHaD from a lifecourse perspective. Although it is well established that the health of both mothers and fathers impacts later offspring health, there is an increasing need for approaches that prioritise adolescents, the next generation of parents, as potential agents of change where lifestyle behaviours become entrained. Future research can include community workshops to explore the level of awareness of adolescence as catalysts for a better future. This will promote conversation about health and well-being and empower the adolescents themselves to be the catalysts of change in their families, schools, and community.

## Figures and Tables

**Figure 1 ijerph-19-06024-f001:**
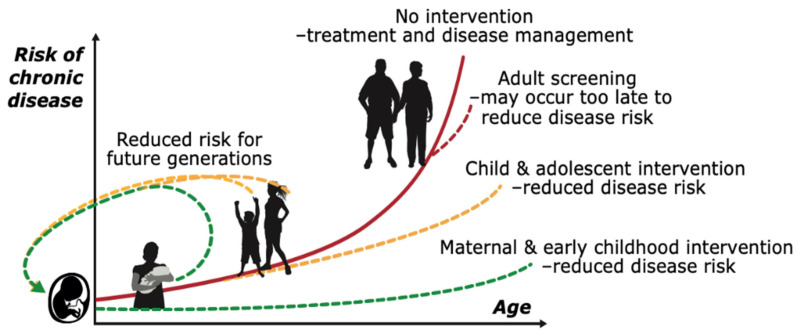
The power of early intervention. Developmental origins of metabolic disease: lifecourse and intergenerational perspectives. The lifecourse approach emphasizes the complex interactions between environmental exposures from pre-conception onwards that influence chronic disease risk. NCD risk increases throughout the lifecourse and as such NCDs do not fit the typical disease model in which an individual is healthy until they contract a disease. The greatest period of developmental plasticity and therefore responsiveness to interventions is in early life. As ageing occurs, responses to new challenges become increasingly inadequate. As such interventions are best timed in early life—although these require a long-term commitment, they are likely to return greater impacts on NCD prevalence than treatment of disease once already manifest. Such an approach also helps mitigate the potential for transgenerational transmission of disease traits. Although an initial focus was on mothers and infants, and more recently on paternal health, there is an increasing recognition of adolescents, as the next generation of parents, as a key lifecourse window of opportunity to break the DOHaD cycle. Adapted with permission from Godfrey et al. [21].

**Figure 2 ijerph-19-06024-f002:**
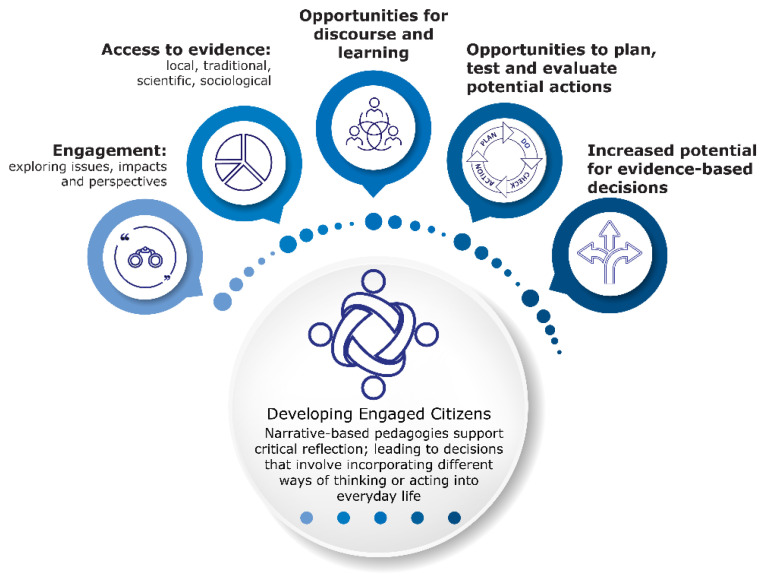
Use of narrative-based pedagogies to support evidence-based actions. Information adapted from Bay et al. [83].

**Figure 3 ijerph-19-06024-f003:**
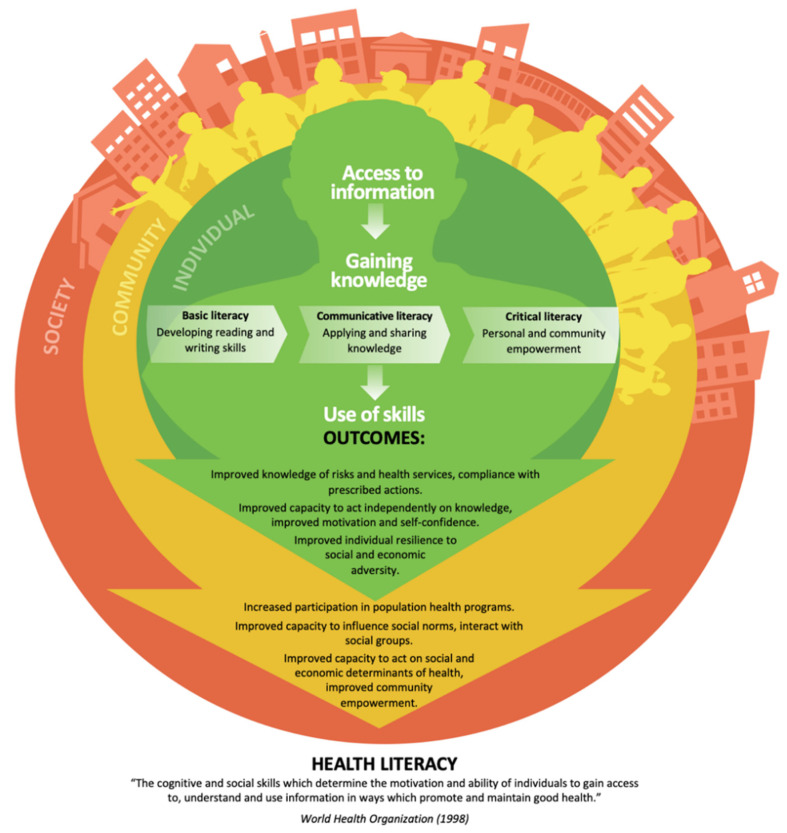
The different levels of Health Literacy. Figure developed using information adapted from Nutbeam [120].

## Data Availability

Not applicable.

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
