# Peer review of "The Developmental Origins of Health and Disease: Adolescence as a Critical Lifecourse Period to Break the Transgenerational Cycle of NCDs—A Narrative Review"

_ijerph, 2022, doi:10.3390/ijerph19106024_

Round 1
Reviewer 1 Report
I am glad having an oppurtunity to review the manuscript entitled " The Developmental Origins of Health and Disease: Adolescence as a critical lifecourse period to break the transgenerational cycle of NCDs - A Narrative Review".
It is a great paper! I particularly appreciate the main idea of this paper that prioritises adolescents, the next generation of parents, as the most important link in development of the origins of health and non-communicable diseases (NCDs).
The thoughts outlined in this article took me back to the eighth and ninth decades of the last century. Although the paper is written based on new references, but the statements are the same as for several decades. I am very surprised that there has been no progress in development of the NCDs control and treatment paradigm so far.
The evidence for priority primary prevention strategies against NCD treatment was highlighted in the Ottawa Charter for Health Promotion (1986). Health promotion movement has become a complementary framework to the traditional focus on health protection and disease prevention. Ottawa Charter has created the vision by clarifying the concept of health promotion, highlighting the conditions and resources required for health and identifying key actions and basic strategies to pursue the WHO policy of Health for All. It highlighted the role of organisations, systems and communities, as well as individual behaviours and capacities in creating opportunities and choices for better health. The World Health Organization has organized, in partnership with national governments and associations, a series of follow up conferences, which have focused on each of Ottawa’s five health promotions strategies. The CINDI (Countrywide Integrated Non-Communicable Diseases Intervention) programme was implemented in many European countries. The focus has been also done on the younger generations as it was noted that NCD has roots in childhood.
I am writing all this because the article lacks the historical perspective of the problem under analysis. The authors should also note what major advances in NCD management and control have been made since the Ottawa Charter in 1986. What is the novelty of this narrative review?
While reading the submitted manuscript, several questions arose and inaccuracies were also noticed, which I recommend to fix.
- Page 1: The reference #1 is not appropriate because it does not contain original data on mortality from NCDs. Data from WHO databases, such as reference #38, would be more appropriate here.
- Section 2: Only the peculiarities of the mother's diet are discussed here. Other risk factors, such as maternal smoking and alcohol consumption, also need to be given more attention.
- Section 4: The importance of health promotion rather than health education needs to be emphasized in this section. The health promotion approach is much more effective than the health education approach.
- Section 6: Don't the authors think that the term "systems" should be replaced by "integrated", or at least used as synonyms?
- References: References must be provided in accordance with the requirements of the journal.
Thank you for considering my opinion. I encourage authors to keep on working to improve the manuscript.
Reviewer 2 Report
The Developmental Origins of Health and Disease: Adolescence as a critical life course period to break the transgenerational cycle of NCDs - A Narrative Review
We would like to congratulate the authors for such a comprehensive overview of lifetime critical periods and NCD. The narrative review by Tohi et al. highlights the importance of adolescence as a critical period of intervention to prevent NCD. It nicely provides concrete examples of Knowledge Translation initiatives during adolescence that may reduce the burden of NCD. To further strengthen the article, we have some suggestions which are listed below.
- The title of the review is NCDs – however, the review focuses mainly on cardio-metabolic disease and diabetes. It may be beneficial to include some other conditions like asthma or respiratory health outcomes.
- In January 2020, Lancet announced a Lancet campaign on child and adolescent health. There are key publications on adolescent health that may be worth citing.
- The authors mention examples of some intervention strategies during adolescence. Are there studies associating those strategies with health outcomes? If yes, can the authors quantify how much is the reduced risk from the intervention strategies mentioned in this review?
- We like that the review includes Figure 1 from Godfrey et al. That figure provides powerful messages, and it will be helpful to either have a more informative figure caption or the message from Figure 1 be better integrated into the article body.
- The system approach section might need some more clarification. What about health literacy in schools or communities-based organizations? Is there evidence to suggest that is absent or lacking? Can the authors reference it back to Figure 3?
- Have a clear objective statement for the review article in the abstract and introduction.
- Figure captions need to be consistent (eg. adapted from [52] vs. Godfrey et al., [13]).
Reviewer 3 Report
This review manuscript describes DOHaD concept and importance of the awareness of transgenerational cycle of NCDs in father and the adolescent. The authors put particular emphasis on the latter.
No one will not disagree with the concept introduced in this review --- adolescence is an important period for breaking the transgenerational cycle of diseases --- and authors’ suggestions that knowledge translation to be undertaken to the adolescent for the awareness that they are responsible to the health of next generation. However, the important is how to effectively forward the KT to actually making the adolescent move to that direction.
Public health has been advocating the awareness of healthy lifestyle to the people of all ages, including adolescent, although it might be only for maintaining the health of their own. However, it has not been successful for some people and unhealthy lifestyle persists in such people. It is indeed these adolescent people whom DOHaD-KT targets to change the attitude toward health: not only KT but some more motivation seems necessary for the change.
It would be practically more informative review if the authors introduce the contents of KTs that made adolescence people aware and actually move toward to breaking transgenerational cycle. Or suggesting the authors’ own idea for it would be fine.
There are several typos in the manuscript that must be corrected.
Round 2
Reviewer 1 Report
The new version of the article has been revised in the light of my comments and authors took them into account as well. I therefore recommend publishing this article.